# The antimicrobial activity of an antiseptic soap against *Candida Albicans* and *Streptococcus Mutans* single and dual-species biofilms on denture base and reline acrylic resins

Camilla Olga Tasso, Beatriz Ribeiro Ribas, Túlio Morandin Ferrisse, Jonatas Silva de Oliveira, Janaina Habib Jorge *

Department of Dental Materials and Prosthodontics, São Paulo State University (Unesp), School of Dentistry, Araraquara, São Paulo, Brazil

* habib.jorge@unesp.br

**Data Availability Statement:** All relevant data are within the manuscript and its Supporting Information files.

## Abstract

To evaluate the effect of antiseptic soap on single and dual-species biofilms of *Candida albicans* and *Streptococcus mutans* on denture base and reline resins. Samples of the resins were distributed into groups (n = 9) according to the prevention or disinfection protocols. In the prevention protocol, samples were immersed in the solutions (Lifebuoy, 0.5% sodium hypochlorite solution and PBS) for 7, 14 and 28 days before the single and dual-species biofilms formation. Overnight denture disinfection was simulated. In the disinfection protocol, samples were immersed in the same solutions during 8 hours after the single and dual-species biofilms formation. Antimicrobial activity was analyzed by counting colony-forming units (CFU/mL) and evaluating cell metabolism. Cell viability and protein components of the biofilm matrix were evaluated using confocal laser scanning microscopy (CLSM). Data were submitted to ANOVA, followed by Tukey's post-test (α = 0.05) or Dunnett's T3 multiple comparisons test. In the prevention protocol, Lifebuoy solution effectively reduced the number of CFU/mL of both species. In addition, the solution decreased the cell metabolism of the microorganisms. Regarding disinfection protocol, the Lifebuoy solution was able of reduce approximately of 2–3 logs for all the biofilms on the denture base and reline resin. Cellular metabolism was also reduced. The images obtained with CLSM corroborate these results. Lifebuoy solution was effective in reducing single and dual-species biofilms on denture base and reline resins.

## Introduction

Denture Stomatitis (DS) is an erythematous chronic inflammatory disease which is commonly encountered in the oral mucous areas that are in direct contact with the dental prosthesis [1]. Its etiology is still controversial and is considered multifactorial, with risk factors such as the presence of microbial infection (mainly *Candida* spp.), poor hygiene of the dentures, and

**Funding:** The author(s) received no specific funding for this work.

**Competing interests:** The authors have declared that no competing interests exist.

trauma due to ill-fitting dentures [2–4]. The incidence of DS is approximately two-thirds of the removable denture wearers and, despite the great number of patients with this condition, most of the time the symptoms are inexistent. Some patients can report a discomfort such as pain or burning sensation [5].

Poor denture cleaning is often observed and is directly associated with a greater biofilm adherence on the denture surface, contributing to the development of DS [6]. The main components of these biofilms are bacteria, with high prevalence of *S. mutans*, and yeasts, highlighting *Candida* species, primarily *C. albicans* [7, 8]. The interaction between *C. albicans* and *S. mutans* is of the commensalism type, when microorganisms can provide substrates and/or metabolites one to other, thus stimulating the growth of both species. The fungus metabolizes sucrose inefficiently but benefits from sucrose hydrolysis products (glucose and fructose) performed by *S. mutans*. The presence of *C. albicans* in the biofilm increases the formation of exopolysaccharides (EPS), the glucans, increasing bacterial adhesion and cohesion. It also forms a diffusion-limiting matrix that protects bacteria and helps to acidify the local microenvironment. Thus, the accumulation and formation of microcolonies of *S. mutans* becomes greater [9–11].

There are several treatments proposed for DS. However, prevention is considered the most effective method to be taken against problems related to the use of dentures. Among prevention methods, overnight removal and immersing dentures in disinfectant solutions is commonly used. In a recently published literature review, it was concluded that overnight storage of dentures in disinfection solution can reduce the *C. albicans* colonization. In addition, immersion can change the surface of the resin and reduce the adhesion of microorganisms [12]. Many disinfectant solutions are reported in the literature, such as sodium hypochlorite, in different concentrations, alkaline peroxide, soap solutions, chlorhexidine and vinegar [13–15]. Sodium hypochlorite is well accepted as a standard disinfectant; however, it can cause allergic reactions, changes in the physical properties of acrylic resins and has corrosive nature while in contact with metals [16, 17].

The soap solution can be an alternative for cleaning removable dentures, considering its effectiveness in reducing biofilm and the absence of cytotoxicity, as well as the unaltered physical and mechanical properties of the acrylic resins after immersion in this solution [14, 18–21]. In additional, it had a good acceptance by denture wearers [18]. Zoccolotti et al. (2018) [14] observed that some disinfectant liquid soaps were effective in reducing the *C. albicans* simple biofilm formed on the surface of acrylic resin samples. Furthermore, the same study concluded that the soaps tested were not toxic and did not cause changes in the roughness of the samples. Considering the promising laboratory results, Tasso et al. (2020) [18] carried out a controlled clinical study with the objective of evaluating the effectiveness of two disinfectant liquid soap solutions (Dettol and Lifebuoy®) in controlling biofilm present in removable complete dentures. The authors concluded that the Lifebuoy® and Dettol solutions had similar results and were effective in reducing biofilm. In another study, it was observed that immersing reline resin in Lifebuoy® soap reduced the biofilm formed on the samples and the immersion did not affect the cytotoxicity of the material [19]. These studies evaluated biological properties in relation to *C. albicans* strains and clinical isolates of *Candida* species. No study was found in the literature evaluating the effect of disinfectant soaps on dual-species biofilms of *C. albicans* and *S. mutans*, which demonstrates the importance of the present study in developing an accessible and low-cost disinfection protocol for wearers of dentures.

Therefore, the objective of this study was to evaluate the effect of disinfectant liquid soap solution, on single and dual-species biofilms (*C. albicans* and *S. mutans*) formed on denture base and reline acrylic resins. Two protocols were performed: 1. Prevention protocol, in which the samples were immersed at different times in the evaluated solutions, and the biofilm was

formed after each time, and then we analyzed the influence of these solutions; 2. Disinfection protocol, in which the ability to remove biofilms formed on the samples was evaluated. The null hypothesis of this study is that the prevention and disinfection protocols will not influence the simple and dual-species biofilms of *C. albicans* and *S. mutans* on the denture base and reline acrylic resin samples.

## Material and methods

### Acrylic resins samples

Denture base acrylic resin (Vipi Wave) and hard chairside relining (New Truliner—Brosworth) samples (14 mm × 1.2 mm) were prepared using a steel mold containing glass plates sandblasted with aluminum oxide to standardize the surface roughness of the samples in, approximately, 3,0 μm [20–22]. The acrylic resin was manipulated according to the manufacturer's recommendations. After polymerization, excess from the samples was removed using a trimming bur and the samples were immersed in distilled water for 48 hours (zero time) at 37˚C to remove residual monomer. After the preparation, the samples were washed in ultrasonic cleaner with distilled water for 15 minutes. Next, surface roughness was measured. All samples with a surface roughness value varying between 2.7 and 3.7 μm [23] were selected for the microbiological tests to simulate the internal surface of the denture.

### Groups

The denture base acrylic resin and hard chairside relining samples were distributed into three groups (n = 9) according to the disinfection solutions: Liquid soap solution at 0.78% [14]; 0.5% sodium hypochlorite solution (positive control) and Phosphate-buffered saline (PBS) (negative control). To obtain the solution, the soap was diluted in saline solution at their appropriate minimum inhibitory concentrations determined in a previous study [14, 18]. The n value was estimated based on previous studies [19–24].

### Prevention protocol

The samples were immersed in the solutions for 0, 7, 14 and 28 days before the single and dual-species biofilms formation (Fig 1). The solutions were changed every day. Overnight denture disinfection was simulated. Thus, the samples remained in the solutions for 8 hours and in distilled water for the other 16 hours, which simulated the period of use of the denture.

To assess the biofilm formation capacity after disinfection, all the samples were firstly immersed in disinfectant solutions in different periods. After, the microorganisms (*C. albicans* ATCC 90028 and *S. mutans* UA 159) were reactivated, and the pre-inoculum and inoculum were prepared to biofilm formation [24]. The tubes containing microorganisms were centrifuged for 5 minutes to form a pellet. After that, the pellet was washed twice with PBS. Next, the microorganisms were resuspended in Roswell Park Memorial Institute (RPMI)-1640 medium. The optical density was measured to achieve a concentration of $1x10^6$ cells/mL [25].

After the immersion periods, the samples were placed in a 24-well plate to form the biofilm. In each well, 750 μL of the microorganism suspension and 750 μL of RPMI-1640 medium were added for the single biofilm. For the dual-species biofilm groups, 750 μL of each microorganism suspension was added. Then, the plates were incubated for 90 minutes (adhesion phase) at 37˚C with 5% CO2.

Following the adhesion phase, the wells were washed twice with PBS, and 1500 μL of fresh RPMI medium were added. The plates were incubated under the previous conditions. After 24

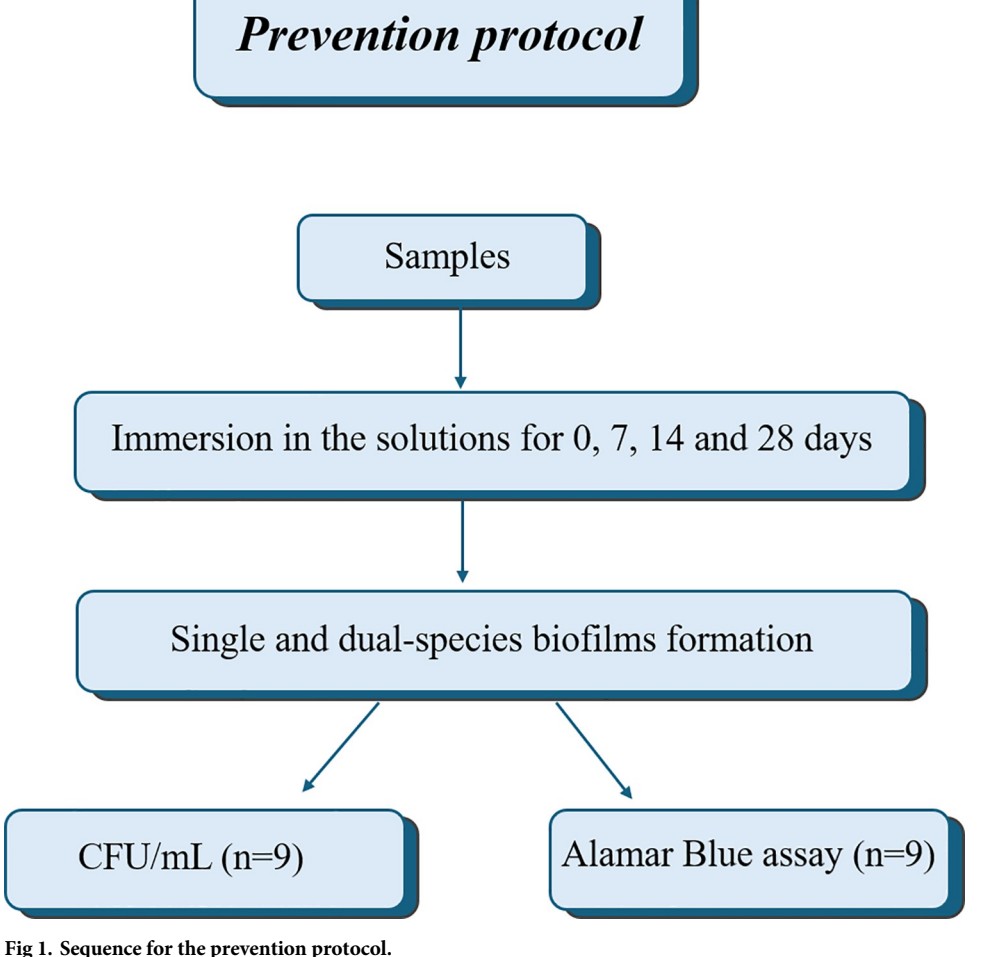

**Fig 1. Sequence for the prevention protocol.**

hours, half of the medium was refreshed, and the plate was incubated for more 24 hours, totalizing 48 hours (mature biofilm).

## Disinfection protocol

The samples were immersed in the solutions for 8 hours after the single and dual-species biofilm formation (Fig 2).

Initially, the samples were washed for 20 minutes in an ultrasonic cleaner and then placed under UV light in the laminar flow cabinet for additional 20 minutes on each side to be disinfected. Similarly to the prevention protocol, the process of biofilm formation was carried out on the samples. Subsequently, following the 48 hours of biofilm maturation, all the medium was removed from each well, and the solutions corresponding to each group were applied (PBS (negative control), 0.5% sodium hypochlorite (positive control), and a Lifebuoy). The specimens remained in contact with the solutions for 8 hours, simulating the denture disinfection process.

## Colony-formation unit assay (CFU/mL)

After completing all the aforementioned procedures, each well was washed twice with PBS, and the specimens were transferred to a new 24-well plate with 150 μL of PBS in each well.

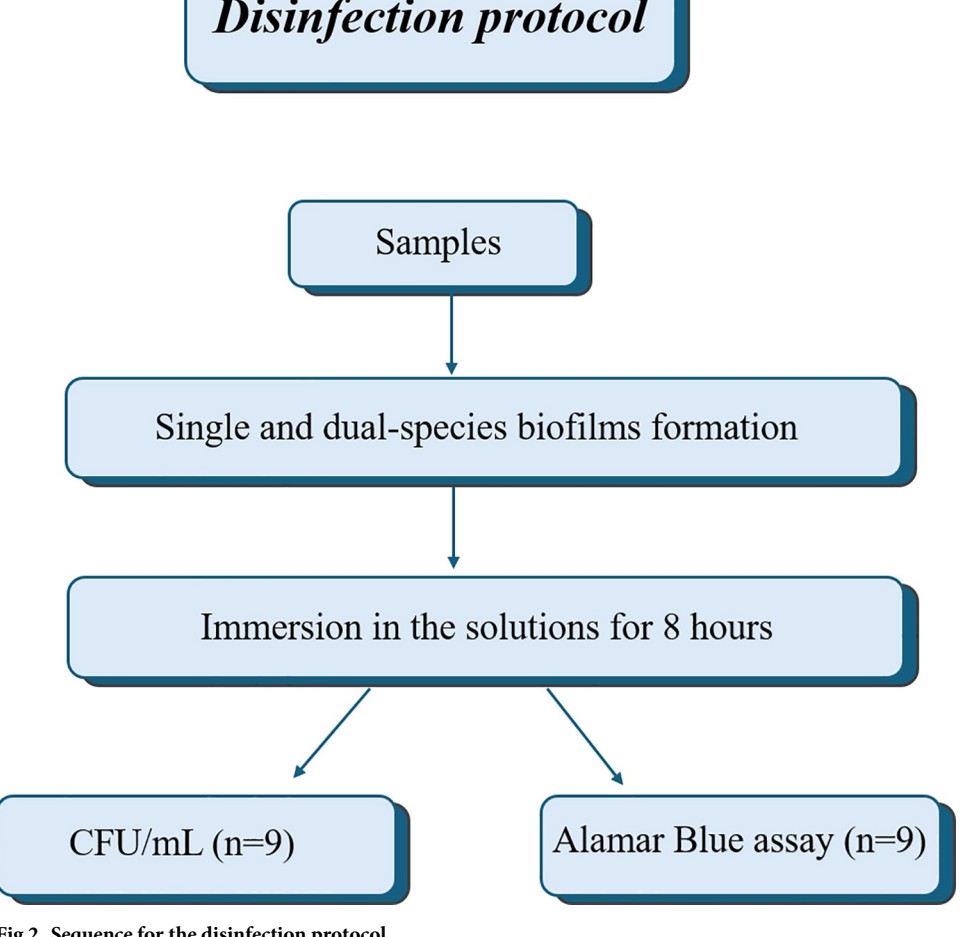

**Fig 2. Sequence for the disinfection protocol.**

Afterward, the samples were scraped using a pipette tip for 1 minute to remove the formed biofilm [20, 21, 24]. Subsequently, serial dilutions ranging from $10^{-1}$ to $10^{-4}$ were prepared, and 25 μL of the $10^{-3}$ and $10^{-4}$ dilutions were plated Sabouraud Dextrose Agar (SDA) and Brain Heart Infusion Agar (BHI) culture medium, supplemented with Amphotericin B, to *C. albicans* and *S. mutans*, respectively. The dishes were then incubated at 37°C with 5% $CO_2$ for 48 hours. Following the incubation period, the number of colony-forming units per milliliter (CFU/mL) was determined [24]. The experiments were performed in triplicate, on three different occasions (n = 9). The experiments were performed by a single experienced operator to ensure a standard protocol.

### Alamar Blue assay

The cell metabolism of *C. albicans*, *S. mutans*, and dual-species biofilms was assessed using the Alamar Blue assay, which measures cell viability by assessing mitochondrial enzyme activity. After immersion in the solutions, the samples were washed twice with PBS and were transferred to a new 24-well plate. Subsequently, 1500 μL of fresh RPMI medium and 150 μL of Alamar Blue solution were added to each well. The plates were then incubated at 37°C with 5% $CO_2$ for 4 hours. After the incubation period, the fluorescence was measured using a

Fluoroskan Ascent at 560 nm (A560) and 590 nm (A590). The experiments were performed in triplicate, on three different occasions (n = 9). The experiments were performed by a single experienced operator to ensure a standard protocol.

## Assessment of cell viability by CLSM

For this assay, after disinfection using the established protocol, the samples (n = 2) were washed twice with PBS and were transferred to a new 24-well plate. Afterward, 10 mL of saline solution containing 2 mL of live/dead dye (SYTO-9 and propidium iodide, PI) from Molecular Probes (Eugene, OR, USA) was added to the samples. The plates containing the specimens were incubated in darkness for 15 minutes to allow for optimal dye penetration and staining. The maximum excitation and emission wavelengths utilized were 480/500 nm for SYTO-9 and 490/635 nm for PI. Images were captured using CLSM (Carl Zeiss LSM 800 with Airyscan), where live cells were stained in green and dead cells appeared in red. A 20× magnification objective was used. For all groups, the fluorescence emission reading was performed separately for dead and live cells [24].

## Assessment of protein components of the biofilm matrix by CLSM

CLSM (Carl Zeiss LSM 800, Airyscan technology, along with the ZEN BLUE 2.3 System Software) was employed for qualitative assessment of protein components within the biofilm matrix. After the disinfection protocol, biofilms were washed twice with 200 μL of PBS buffer per well. Subsequently, 100 μL of SYPRO® Ruby Biofilm Matrix Stain solution (Invitrogen™) was added to each well for staining. The plates were incubated anaerobically at 37˚C for 30 minutes in the dark. Afterward, the stain was then removed from the samples, and each well was rinsed with 100 μL of PBS buffer. Following PBS rinse, the samples were visualized using confocal fluorescence microscopy with 488 nm excitation and 700 nm emission. The biofilms were analyzed using a 10× magnification objective [24].

## Statistical analysis

The collected data were submitted to the Shapiro-Wilk test to assess normality and the F test to assess homoscedasticity. To the prevention protocol, the data from the CFU/mL assay were submitted to three-way ANOVA (time, solution, and biofilm type) and data from Alamar Blue assay were submitted to two-way ANOVA (time, solution), followed by Tukey's post-test. Considering the disinfection protocol, the data were submitted to two-way ANOVA, followed by Tukey's post- test. The post-hoc adjustment for multiple comparisons was made only for Alamar blue assay in disinfection protocol (Dunnett's T3 and Dunn' multiple comparisons tests). For all other statistical analysis, the multiple comparisons were performed by mean ± 95% confidence interval estimation and the followed results illustrated in graphs. In this case, if the space between the error bar did not match there is a significant statistical difference among the groups evaluated. All statistical analyses were performed with a significance level of 5% using SPSS software (v21.0).

## Results

### Prevention protocol

The CFU/mL count (logarithmic scale) underwent assessment through a 3-way analysis of variance, considering factors such as time, biofilm type, and solution. The analysis of CFU/mL of *C. albicans* on the denture base and reline acrylic resins samples revealed a significant

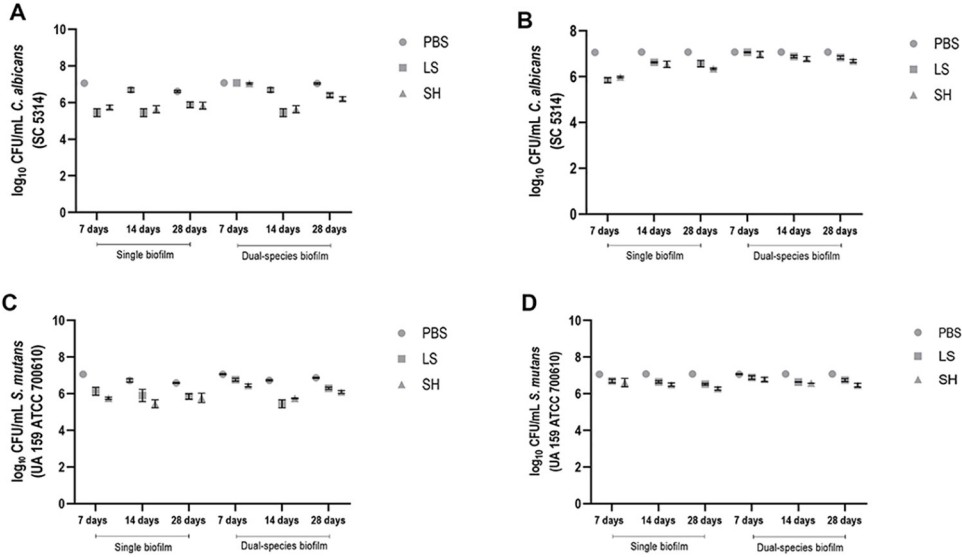

**Fig 3.** CFU/mL values of *C. albicans* and *S mutans* cells (single and dual-species biofilms) referring to biofilm formation on denture base (A and C) and reline acrylic resin (B and D) samples after immersion in PBS, Lifebuoy, and 0.5% sodium hypochlorite solutions.

interaction p<0.0001 between immersion time and the type of biofilm (single and dual-species), as depicted in Fig 3.

Fig 3A and 3B illustrate the viability values of *C. albicans* (log10 CFU/mL) on the denture base and reline acrylic resins samples, respectively. The data showed no statistical difference between the sodium hypochlorite and Lifebuoy groups regarding biofilm formation at all time points. Notably, both these groups demonstrated significant differences from the PBS group, except on the 7th day in the dual-species biofilm. These findings highlight the effectiveness of Lifebuoy soap, which exhibited comparable results to the positive control group (0.5% sodium hypochlorite).

Regarding *S. mutans* growth in single and dual-species biofilms on the denture base acrylic resin samples after immersion in different solutions (Fig 3C), there was a notable reduction in microbial viability for the Lifebuoy group after 14 days, regardless of the biofilm type. At all-time points, there was no statistically significant difference between the Lifebuoy and hypochlorite groups, except on the 7th day of immersion for both biofilms, indicating the effectiveness of the soap against *S. mutans*.

For the denture relining samples, the hypochlorite group demonstrated a greater reduction in microbial viability for the single biofilm of *S. mutans* on the 28th day of immersion (Fig 3D). This group exhibited statistical difference from all other groups across all evaluated conditions. Additionally, the hypochlorite group showed a statistically significant difference from the Lifebuoy group for both single and dual-species biofilms of *S. mutans* during the same immersion period. Notably, for most immersion time periods, there was a significant difference between the PBS and Lifebuoy groups, with the Lifebuoy group exhibiting a greater reduction in cell viability.

The Fig 4A to 4C shows the mean of fluorescence values corresponding to the cell metabolism of biofilm formed on denture base acrylic resin samples (Alamar Blue assay). By considering the data on cell metabolism of *C. albicans* in single biofilm on denture base acrylic resin samples (Fig 4A), a significant interaction between the factors was observed (p<0.0001). The Lifebuoy and sodium hypochlorite groups exhibited a significant difference compared to the

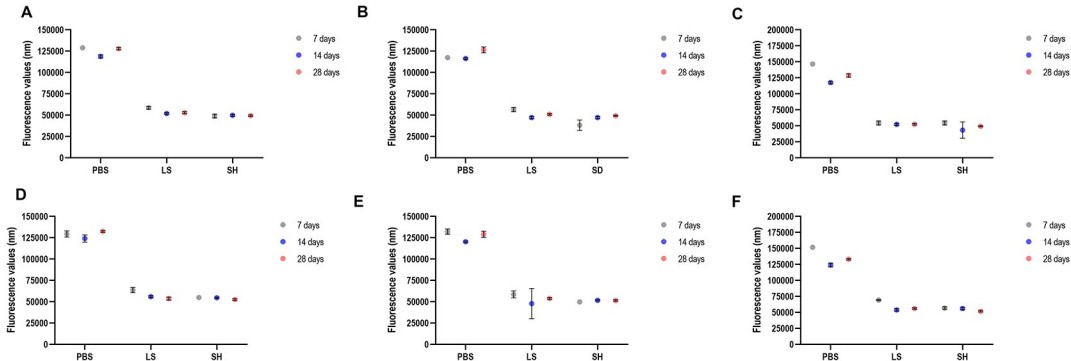

**Fig 4.** Results of the cellular metabolism of the *C. albicans* and *S. mutans* cells (single and dual-species biofilms) referring to biofilm formation on denture base (A-C) and reline acrylic resin (D-F) samples after immersion in PBS, Lifebuoy, and 0.5% sodium hypochlorite solutions.

PBS group across all immersion periods. Additionally, the groups in which the samples remained immersed in the Lifebuoy solution for 14 and 28 days showed statistical similarity to the sodium hypochlorite groups after 7, 14, and 28 days of immersion.

Concerning *S. mutans* single biofilm on denture base acrylic resin samples (Fig 4B), the group with the greater microbial reduction was the 7-day immersion in sodium hypochlorite, being statistically different to the other groups. Additionally, microbial reduction in the 14 and 28-day immersion in Lifebuoy groups was statistically similar to hypochlorite group, both reduced the cell metabolism. The results of the dual-species biofilm on denture base acrylic resin samples showed that the experimental group (Lifebuoy) and the sodium hypochlorite group presented the greater metabolism reduction when compared to PBS group (Fig 4C).

When analyzing the results of cell metabolism on the relining specimens (Fig 4D to 4F), a significant interaction between the independent factors was observed ($p < 0.0001$). In the case of the *C. albicans* single biofilm (Fig 4D), both the Lifebuoy and sodium hypochlorite groups exhibited a significant difference compared to the PBS group. Regarding the immersion time, the experimental group (Lifebuoy) after 14 and 28-day immersion showed statistical similarity to the sodium hypochlorite group after 7, 14, and 28 days of immersion. A significant reduction in cell metabolism was observed in the *S. mutans* single biofilm for both the Lifebuoy and sodium hypochlorite groups compared to the PBS group (Fig 4D). Additionally, no significant difference was found in relation to the immersion time. The same groups presented similar results for the dual-species biofilm at all time points (Fig 4F). However, in terms of time, the Lifebuoy group after 14 and 28-day of immersion showed statistical similarity to the sodium hypochlorite group after 7, 14, and 28 days of immersion. This significant interaction is also observed in the PBS group at all time points, where each immersion period is significantly different from the others.

## Disinfection protocol

About CFU/mL, there was a significant difference between the factors in all statistical analyses ($p < 0.0001$). The Lifebuoy solution exhibited a nearly 3-log reduction in biofilm viability on the denture base acrylic resin compared to the negative control across all types of biofilms. As for the relining samples, there was an approximate 2-log reduction in all biofilms (Fig 5).

It was demonstrated that the sodium hypochlorite solution completely reduced the CFU/mL values of *C. albicans* and *S. mutans* single biofilm in both resins (Fig 5A and 5B). Moreover, the Lifebuoy group exhibited a statistical difference compared to the PBS group, thereby

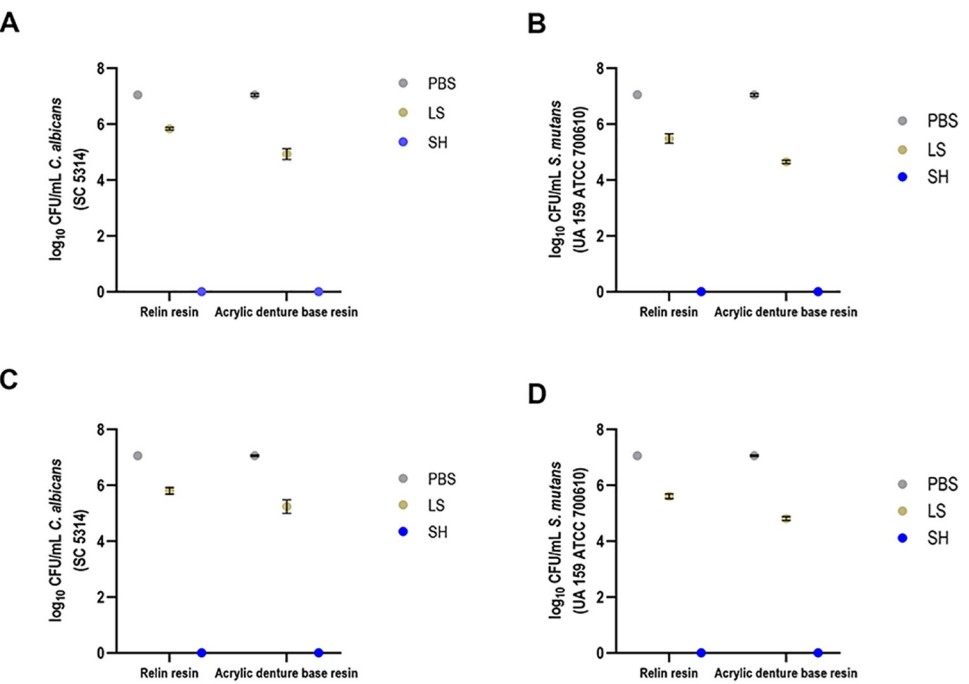

**Fig 5.** CFU/mL values of single (A and B) and dual-species (C and D) biofilms formed on the denture base and reline acrylic resin samples, submitted to the disinfection procedure.

indicating the effectiveness of the experimental solution. Concerning *S. mutans* single biofilm (Fig 5B), the reduction was greater for the denture base acrylic resin.

Regarding the dual-species biofilm (Fig 5C and 5D), the sodium hypochlorite solution completely reduced the CFU/mL values of both *C. albicans* and *S. mutans*, while the Lifebuoy group showed a statistical difference compared to the PBS group. A more significant reduction was observed in the denture base acrylic resin.

Fig 6A to 6C presents the outcomes concerning the cellular metabolism (Alamar Blue assay) of single and dual-species *C. albicans* and *S. mutans* biofilm formed on denture base resin samples. Analyzing the findings, it was possible to observe a significant interaction between the factors ($p < 0.0001$), which leads to the execution of Dunnett's T3 multiple comparisons test. Among all tested groups there was notably difference between both PBS and Lifebuoy liquid soap ($p < 0.0001$), as well as PBS and 0.5% sodium hypochlorite ($p < 0.0001$). However, there was no difference in the interaction between the positive control and the experimental group. Regarding the cellular metabolism of the biofilm formed on hard relining samples (Fig 6D to 6F) it was noted a significant interaction between the groups ($p < 0.0001$). The results were like those aforementioned, no statistical difference was observed between sodium hypochlorite and the experimental group. Additionally, both these groups exhibited a significant contrast with the negative control group ($p < 0.0001$).

Figs 7 and 8 presents images (CLSM) of biofilms grown on both denture base and reline acrylic resin specimens, respectively. For both resins, in the negative control group (PBS) a large number of cells stained in green were observed. On the other hand, the hypochlorite solution (positive group) eliminated both viable and non-viable cells. Similarly, the disinfectant soap solution (experimental group) reduced the number of biofilm cells. This reduction was observed for both single and dual-species biofilms of *C. albicans* and *S. mutans*.

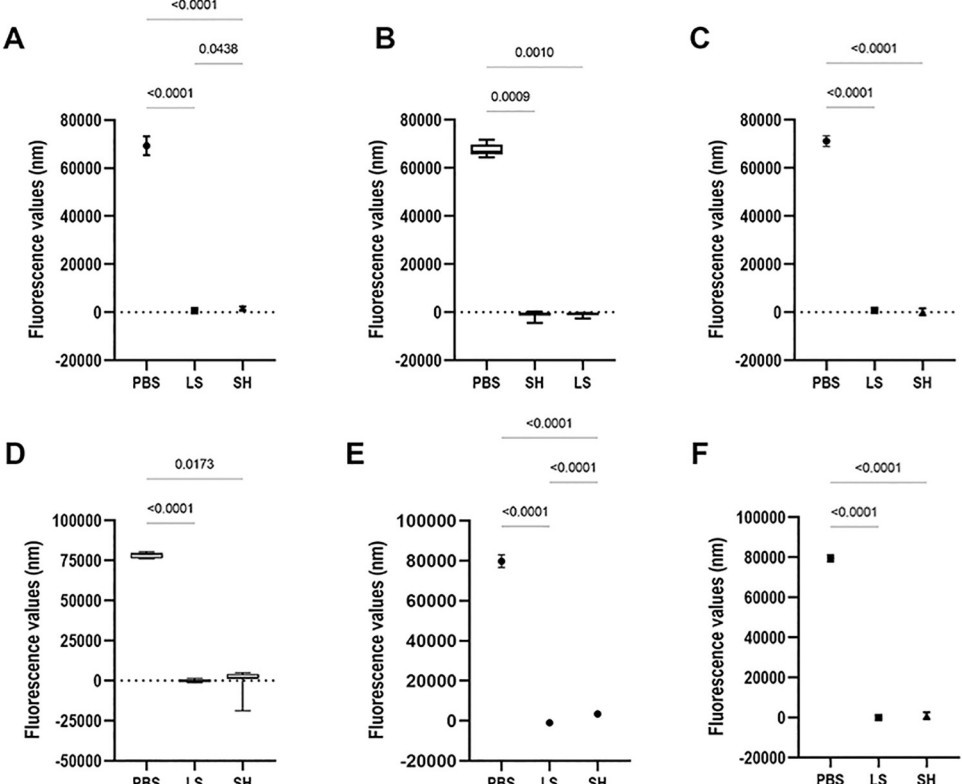

**Fig 6.** Results of the cellular metabolism of the *C. albicans* and *S. mutans* cells (single and dual-species biofilms) formed on denture base (A-C) and reline acrylic resins (D-F) samples.

Regarding the protein components of the biofilm matrix, it was observed that the biofilms treated with hypochlorite solution (positive group) and disinfectant soap solution (experimental group) exhibited a lower level of fluorescence compared to the PBS (negative control group) (Figs 9 and 10). This reduction was observed for both single and dual-species biofilms of *C. albicans* and *S. mutans* formed on both resins, denture base and reline acrylic resin.

## Discussion

The need for an alternative disinfection protocol, equally effective and without any undesirable surface property alterations or allergic reactions, serves as the motivation for this study. In addition, in order to propose effective protocols for the treatment and prevention of denture stomatitis, it is essential to understand the pathogenic microbial species involved in biofilm formation [26].

According to the literature, it was identified that both the *S. mutans* bacteria and the *C. albicans* fungus play a relevant role in the formation of the biofilm on the inside of the denture [7, 24, 27]. Furthermore, it was found that *S. mutans* is an important precursor of biofilm formation and, in association with *C. albicans*, can cause more virulent denture stomatitis [28]. *S. mutans* can easily adhere to the hyphae of *C. albicans*, and the two species can synergistically grow in the biofilm, increasing its thickness, which causes greater damage and makes it difficult to remove or penetrate drugs [29–31]. Therefore, this study aimed to analyze the effectiveness of the Lifebuoy solution in the prevention and disinfection of single and dual-species biofilms of *C. albicans* and *S. mutans* formed on the denture base and reline acrylic resins

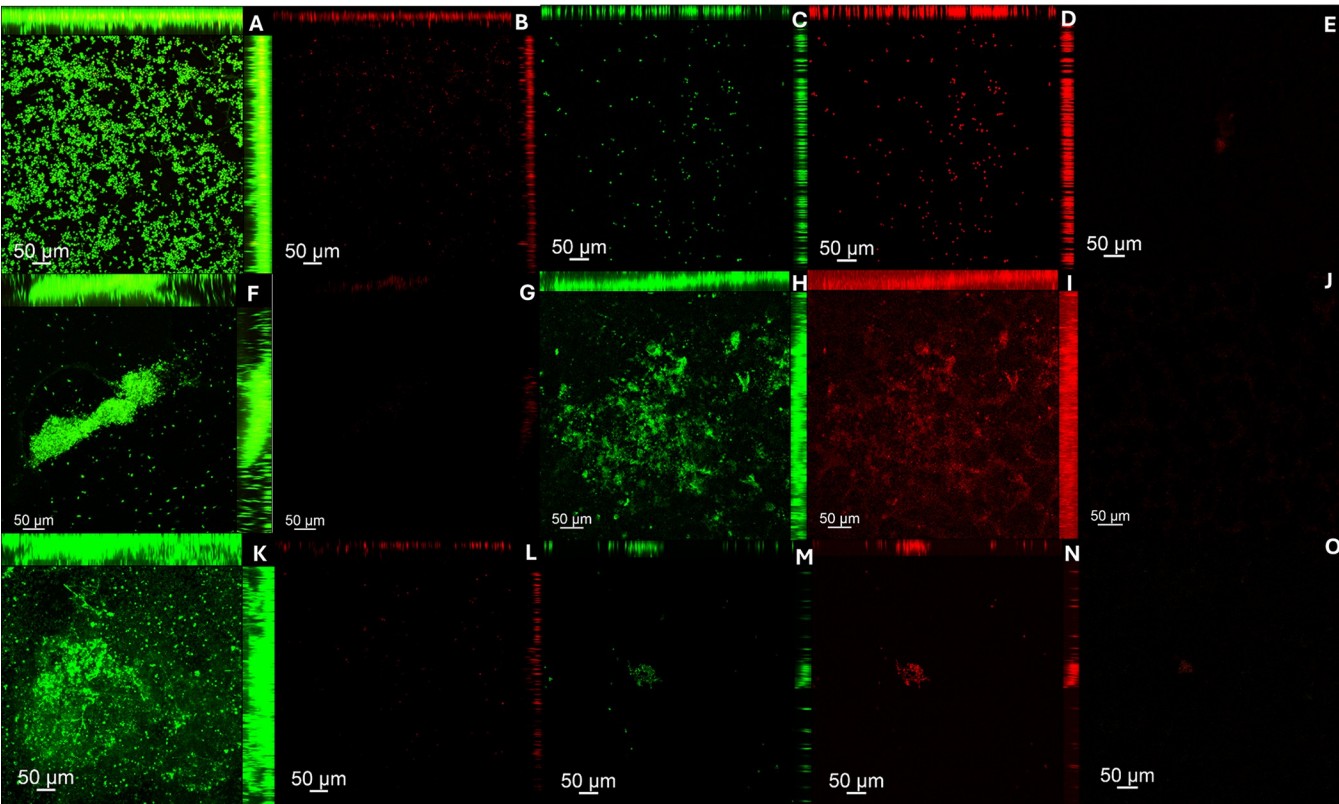

**Fig 7. Cell viability of *C. albicans* and *S. mutans* single and dual-specie biofilms analyzed by CLSM on denture base acrylic resin samples.** Samples labeled with Syto-9 (cells marked in green) and propidium iodide (cells marked in red). (A, B) *C. albicans* biofilm–PBS (Negative control); (C, D) *C. albicans* biofilm–Lifebuoy solution; (E) *C. albicans* biofilm–Sodium Hypochlorite 0,5% (Positive control); (F, G) *S. mutans* biofilm—PBS (Negative control); (H, I) *S. mutans* biofilm—Lifebuoy solution; (J) *S. mutans* biofilm—Sodium Hypochlorite 0,5% (Positive control); (K, L) Dual-species biofilm—PBS (Negative control); (M, N) Dual-species biofilm—Lifebuoy solution; (O) Dual-species biofilm–Sodium Hypochlorite 0,5% (Positive control).

samples. The results indicate an easy and inexpensive method for denture disinfection, particularly for elderly patients with limited manual dexterity. In addition, immersion in disinfectant solutions could be a common protocol in hospitals and nursing homes for denture care among institutionalized patients.

The null hypothesis of this study was rejected as the results demonstrated that immersion in the Lifebuoy solution effectively reduced the formation of both *C. albicans* and *S. mutans* biofilms, whether single or dual-species, formed on the denture base acrylic resin and hard chairside relining samples. The reduction was observed for both the prevention protocol and the disinfection protocol. The application of this alternative immersion protocol could be considered as an option for prevention and disinfection, offering patients a new choice that is cost-effective and easily accessible.

To simulate clinical conditions, various immersion times were evaluated. This is important because the daily immersion of the dentures in disinfectant solutions can cause the incorporation of substances into the acrylic resin, which can cause superficial changes and, consequently, influence the process of adhesion and biofilm formation [12, 14]. Due to the daily exchange of solutions, degradation conditions and release of components can be maintained over time. In this study, there was minimal impact of immersion time on biofilm adhesion and formation.

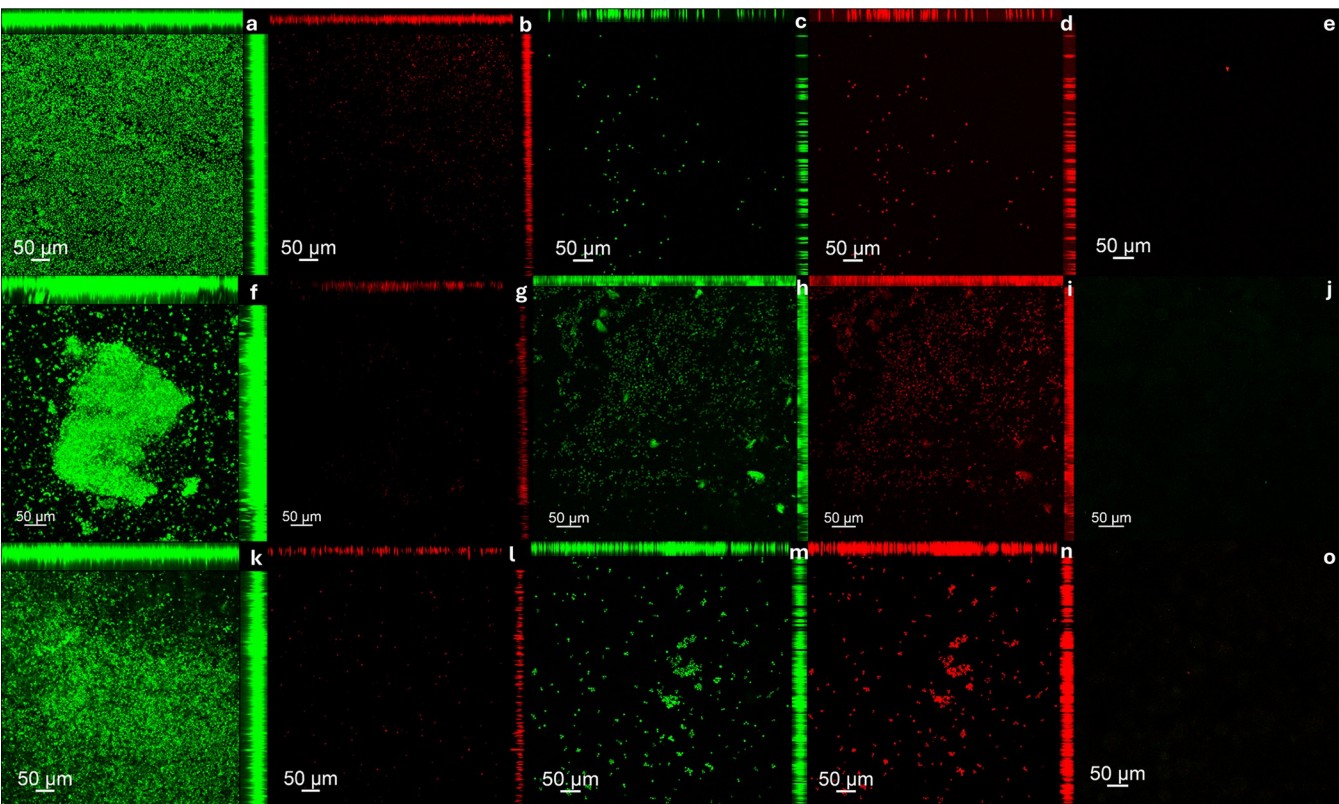

**Fig 8. Cell viability of *C. albicans* and *S. mutans* single and dual-specie biofilms analyzed by CLSM on hard reline acrylic resin samples.** Samples labeled with Syto-9 (cells marked in green) and propidium iodide (cells marked in red). (a, b) *C. albicans* biofilm–PBS (Negative control); (c, d) *C. albicans* biofilm–Lifebuoy solution; (e) *C. albicans* biofilm–Sodium Hypochlorite 0,5% (Positive control); (f, g) *S. mutans* biofilm—PBS (Negative control); (h, i) *S. mutans* biofilm—Lifebuoy solution; (j) *S. mutans* biofilm—Sodium Hypochlorite 0,5% (Positive control); (k, l) Dual-species biofilm—PBS (Negative control); (m, n) Dual-species biofilm–Lifebuoy solution; (o) Dual-species biofilm–Sodium Hypochlorite 0,5% (Positive control).

In the prevention protocol, in which the samples are immersed in the solutions before biofilm formation, it was observed that, for all times, there was no statistical difference between the sodium hypochlorite and Lifebuoy groups in relation to biofilm formation. Furthermore, these two groups were significantly different from the PBS group (negative control), showing the effectiveness of Lifebuoy soap against biofilm formation and in reducing microorganisms equivalent to 0.5% hypochlorite. These results were observed for all types of biofilms and for both resins (denture base and reline, except on the 7th day in the dual-species biofilm). Although statistically different from the PBS control group, the reduction in biofilm in the experimental groups was slight.

This result regarding prevention was compatible with other published study, such as that of Zoccolotti et al., [14] who conducted an in vitro study investigating the ability of *C. albicans* biofilm formation on the denture base acrylic resin samples through immersion in solutions of the antimicrobial liquid soaps most found in pharmacies and markets (Protex, Lifebuoy and Dettol). In the results of Zoccolotti et al., [14] in the prevention protocol, biofilm growth was also observed on the resin samples for the denture base. In both studies, the samples remained immersed in the disinfectant solutions for a period of 8 hours, simulating the overnight protocol, which is recommended because it presents more effective results when compared to shorter periods of immersion and because it encourages the night removal of the prostheses [32, 33]. The reduction of biofilm viability on samples immersed in disinfectant solutions was

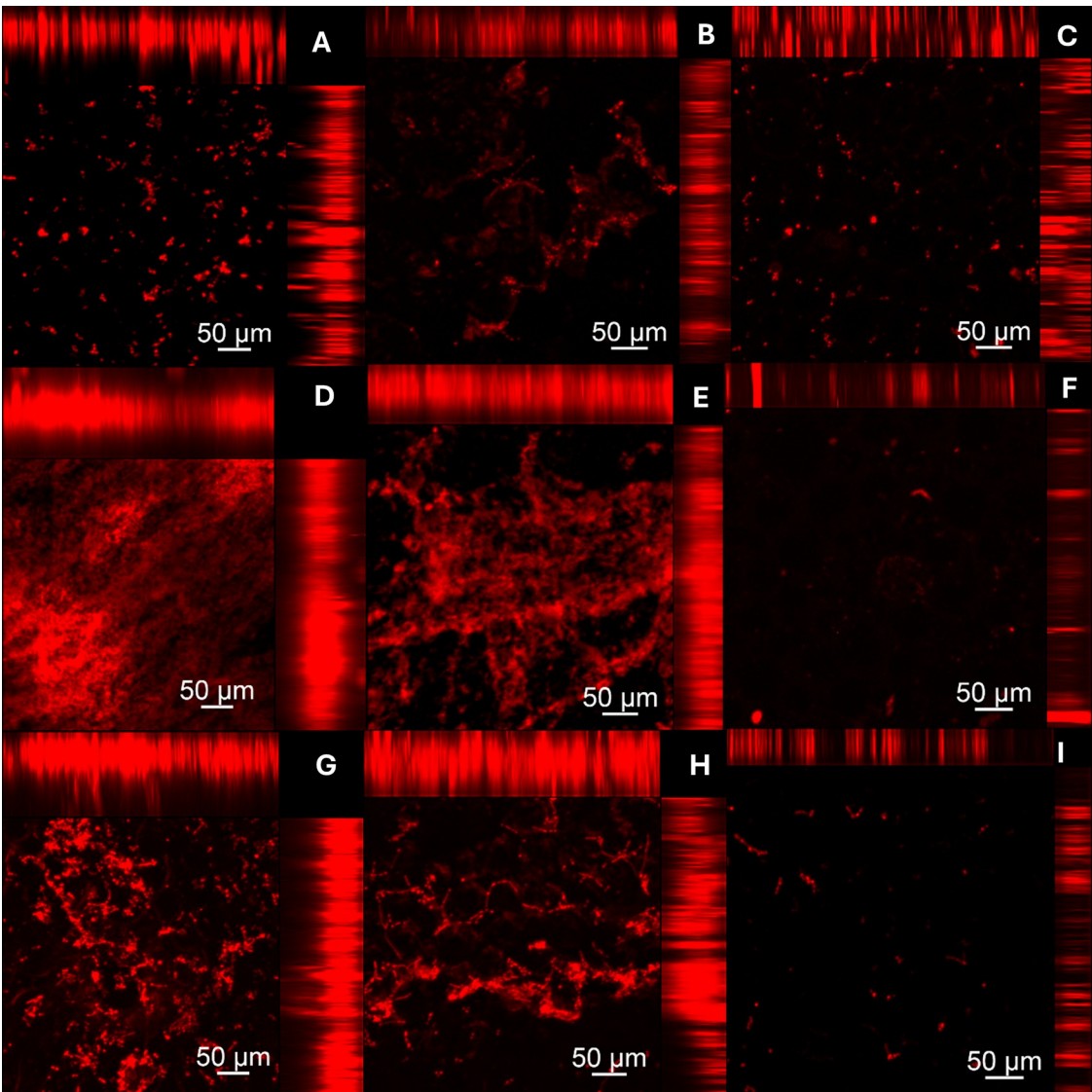

**Fig 9. Assessment of damage to the protein components of the biofilm extracellular matrix after treatment protocols on denture base acrylic resin samples.** Components visualized in red were marked with Sypro Ruby Biofilm Matrix Stain. (A) *C. albicans* biofilm–PBS (Negative control); (B) *C. albicans* biofilm–Lifebuoy solution; (C) *C. albicans* biofilm–Sodium Hypochlorite 0,5% (Positive control); (D) *S. mutans* biofilm—PBS (Negative control); (E) *S. mutans* biofilm—Lifebuoy solution; (F) *S. mutans* biofilm—Sodium Hypochlorite 0,5% (Positive control); (G) Dual-species biofilm—PBS (Negative control); (H) Dual-species biofilm —Lifebuoy solution; (I) Dual-species biofilm–Sodium Hypochlorite 0,5% (Positive control).

observed for both assays, counting colony forming units and the Alamar Blue® test. This means that the disinfectant solutions promoted, in addition to inhibiting cell proliferation, a reduction in their metabolism. It is important to emphasize that no study was found in the literature regarding single biofilm of *S. mutans* and dual-species biofilm of *C. albicans* and *S. mutans* using the prevention protocol with the Lifebuoy soap solution, making it impossible to compare with other results and highlighting the importance of the present study.

Regarding the disinfection protocol of denture base and reline acrylic resins samples, after the formation of biofilms, it was possible to observe decrease in colony forming units in the experimental (Lifebuoy) and the sodium hypochlorite groups compared with the negative

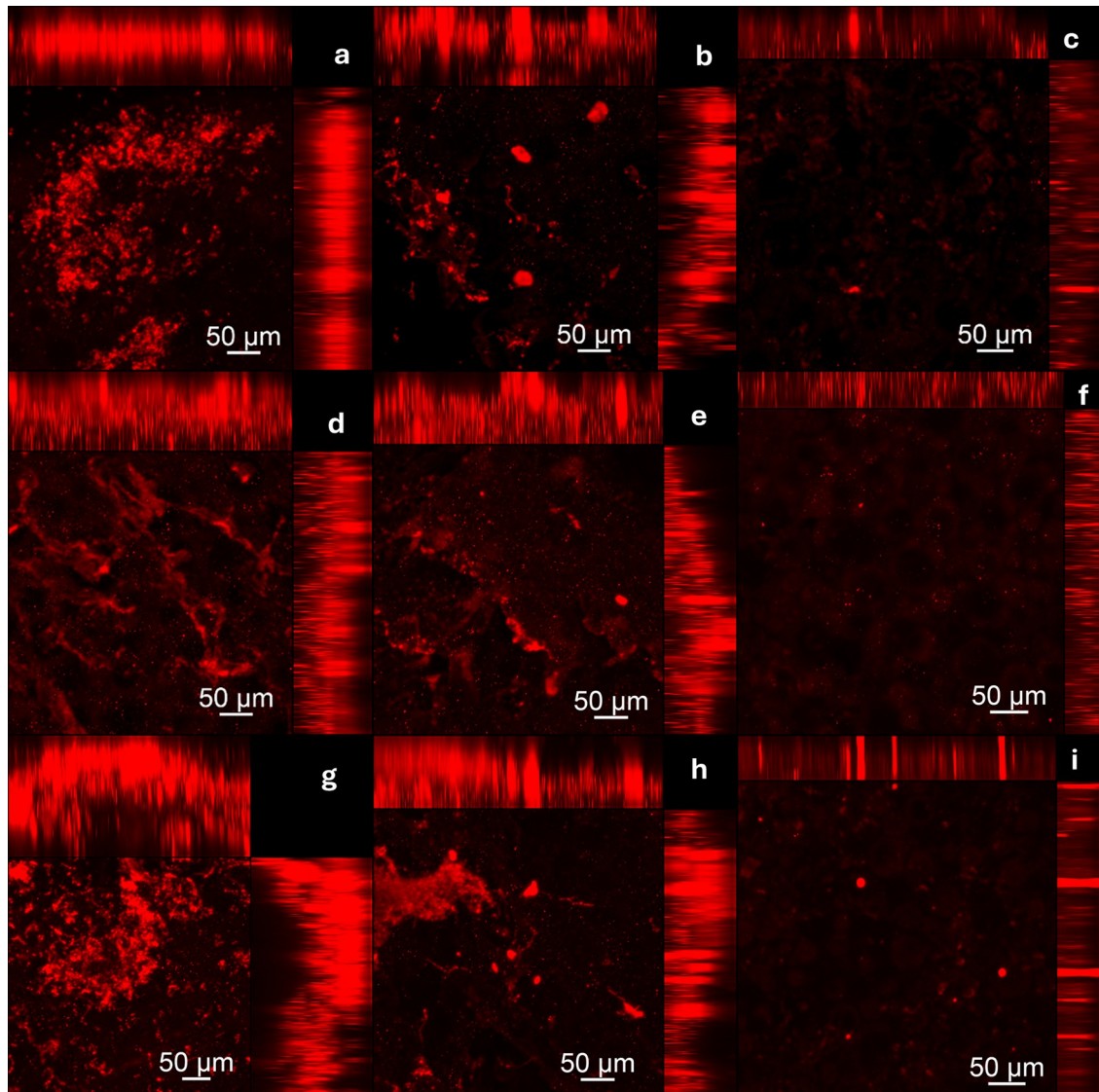

**Fig 10. Assessment of damage to the protein components of the biofilm extracellular matrix after treatment protocols on hard reline acrylic resin samples.** Components visualized in red were marked with Sypro Ruby Biofilm Matrix Stain. (a) *C. albicans* biofilm–PBS (Negative control); (b) *C. albicans* biofilm–Lifebuoy solution; (c) *C. albicans* biofilm–Sodium Hypochlorite 0,5% (Positive control); (d) *S. mutans* biofilm—PBS (Negative control); (e) *S. mutans* biofilm—Lifebuoy solution; (f) *S. mutans* biofilm—Sodium Hypochlorite 0,5% (Positive control); (g) Dual-species biofilm—PBS (Negative control); (h) Dual-species biofilm—Lifebuoy solution; (i) Dual-species biofilm–Sodium Hypochlorite 0,5% (Positive control).

control group (PBS). For the denture base resin samples, for all types of biofilms, there was an average decrease of 3 logs in relation to the control group, while for the reline resin samples this reduction was around 2 logs. This difference between the resins can be attributed to the material's surface characteristics. Relining materials, with their greater surface roughness compared to denture base resins, create a more favorable environment for biofilm accumulation [24]. The Alamar Blue® test showed a reduction in cellular metabolism for the experimental (Lifebuoy) and the sodium hypochlorite groups in relation to the negative control, corroborating the results of the counting colony-forming units. Thereby, the disinfectant solutions promoted, in addition to reducing biofilm formed on the resins, a reduction in their metabolism.

These results also corroborate the study by Zoccolotti et al., [14] who verified that the Lifebuoy solution was able to reduce the simple *C. albicans* biofilm formed on the base resin specimens after 8 hours of immersion. Other studies that reinforce the results obtained are those by Tasso et al., [18] who verified in a clinical study the effectiveness of the Lifebuoy soap solution in disinfecting the biofilm of *Candida* species on complete removable dentures, and by Ribas et al., [20, 21] in which observed a statistically significant reduction in the biofilm of *C. albicans*, formed on both denture base and reline acrylic resins, through brushing and immersion with soap solution. Regarding the results on the reline resin samples, these are compatible with the study published by Malavolta et al., [19] who concluded that the Lifebuoy liquid soap solution was able to reduce the *C. albicans* biofilm on the samples, by around one log in relation to the negative control. As with the prevention protocol, no study was found in the literature regarding single biofilm of *S. mutans* and dual-species biofilm of *C. albicans* and *S. mutans* using the disinfection protocol with Lifebuoy liquid soap solution, making it impossible to comparison with other results and highlighting the importance of the present study.

As positive control, 0.5% sodium hypochlorite was tested in the present study. As in other studies already carried out [34, 35], the effectiveness of this solution was verified in the elimination of microorganisms such as *C. albicans* and *S. mutans* on denture base and reline acrylic resins. Comparatively, it was observed that the Lifebuoy soap also reduced the biofilms in relation to the control group, which points to this disinfection product as an alternative to sodium hypochlorite.

CLSM images support the findings from other assays (CFU/mL and Alamar Blue assay). Regarding to cell viability, the positive control group treated with hypochlorite solution showed eradication of biofilm cells, with no staining observed. So, both viable and non-viable cells were eliminated, in line with other results, highlighting its effectiveness as a disinfectant [36]. Lifebuoy solution also significantly reduced the number of cells. This reduction was observed for both single and dual-species biofilms of *C. albicans* and *S. mutans*, showing the effectiveness of the soap. These findings could be explained by the fact that some antimicrobial substances, in addition to killing the biofilm microorganisms, can act in removing them [37]. The elimination of microorganisms could be related to the change in the pH of the medium. Changes in pH directly alter the surface properties of both microorganisms and solid surfaces. This can involve modifications to the charge and hydrophobicity (water-repelling nature) of their surfaces. Depending on the specific pH shift (acidic or alkaline), the electrostatic repulsion between the microorganisms and the surface can either increase or decrease [38]. The images obtained with SYPRO® Ruby, which specifically targets protein components of the matrix, corroborate these results.

The disinfecting ability of Lifebuoy liquid can be explained by its composition. Among the active principles are: EDTA (ethylenediaminetetraacetic acid), cocamidopropyl betaine, sodium hydroxide, citric acid and triclocarban (which has mechanisms of action very similar to triclosan), curcumin, to name but a few. However, the manufacturer does not mention the concentrations of each component, which makes it difficult to clarify the results obtained [14]. Devine et al. [39] performed a study in which investigated the effectiveness of tetrasodium EDTA, one of the compounds of the Lifebuoy, to disinfect dental products that suffer contamination, such as denture base material and toothbrushes. Effects on *Candida* species were evaluated due to the role of these organisms in denture-associated infections. After treatment lasting 16 hours, tetrasodium EDTA reduced viable counts of salivary bacteria *S. mutans* and *C. albicans* biofilm present on toothbrush bristles by more than 99% and removed most biofilms from the surface of the material. Cocamidopropyl betaine, an analogue to amphoteric detergents, is sourced from extended-chain alkyl betaines. Its reputation for being gentler compared to sodium lauryl sulfate has resulted in its widespread inclusion in numerous

hygiene products [40]. Its primary role involves the breakdown of oils and fats, employing a milder approach than sodium lauryl sulfate. Minnich et al. [41] performed an in vitro study to examine the antimicrobial effects of a solution containing 0.1% betaine. The researchers observed reductions of 5.3 log to 5.8 log for various microorganisms, among them *C. albicans*. Sodium hydroxide can neutralize amino acids and degrade fatty acids, acting on the microorganism cell membrane, which provides its antimicrobial property [42]. The soap solution is additionally formulated with Curcuma Longa. This constituent is recognized for their antimicrobial properties. The biological functions of turmeric curcuminoids are firmly grounded in scientific research, encompassing attributes such as antioxidation, antiinflammation, anticancer properties, antimicrobial effects, neuroprotection, cardio protection, and even radioprotection. These benefits, among others, contribute to the widespread utilization of these natural compounds. Notably, Hsieh et al. [43] have previously established the antifungal impact of curcuminoids on pathogens associated with *C. albicans* growth. This is attributed to the presence of a methoxy group in the compounds, which enhances lipophilicity, facilitating the dissolution of cellular fats and enabling easier penetration of fungal cell membranes, ultimately impeding growth. Li et al. [44] evaluated the influence of curcumin on the *S. mutans* biofilm across different time intervals. The findings indicated that curcumin exhibited dual effects on the viability of the *S. mutans* biofilm, manifesting both in the short-term and over extended periods. Concerning triclocarban (another compound of the Lifebuoy), it was observed that it is a highly effective and broad-spectrum antimicrobial and antiseptic agent [45, 46].

In addition, the soap's mechanisms of action may also be related to the presence of benzalkonium chloride, a quaternary ammonium compound common in non-alcohol hand sanitizers [47]. The positively charged "headgroup" of benzalkonium chloride is attracted to and progressively adsorbs to the negatively charged phosphate heads of phospholipids in the lipid bilayer. This accumulation disrupts the membrane's fluidity, consequently creating hydrophilic gaps within the structure [48]. In addition, the alkyl chain "tail" component of benzalkonium chloride disorganizes the membrane bilayer by inserting itself into the barrier and disrupting its physical and biochemical properties [47, 48].

It is important to emphasize that brushing remains a cornerstone of oral hygiene, and the proposed protocol could offer an additional benefit when used in conjunction with regular brushing. Previous studies were performed and it was concluded that brushing with Lifebuoy soap solution did not change the surface properties of denture base acrylic resin. In addition, the Lifebuoy soap solution was effective in reducing the biofilm formed on the denture base and reline acrylic resins [20, 21].

There are limitations to the present study, such as the complexity of the assessed biofilms and its nature as an *in vitro* study, which limits its applicability to clinical practice. However, taking into account the effectiveness of the antimicrobial action of Lifebuoy soap, the results of the present study are promising. Therefore, due to the aforementioned reasons, further studies with more complex biofilms and studies on possible mechanisms of action are necessary to substantiate the efficacy of the proposed protocol.

## Conclusion

Based on the limitations of the present study, it can be concluded that the use of Lifebuoy soap solution has the potential to reduce the number of colonies forming units and the cellular metabolism of *C. albicans* and *S. mutans* in both single and dual-species biofilms formed on denture base and reline acrylic resins samples. These findings support the effectiveness of Lifebuoy soap as an antimicrobial agent and highlight its cost-effectiveness and accessibility as a viable option for cleaning complete dentures.

## Supporting information

**S1 File.**
(DOCX)

## Author Contributions

**Conceptualization:** Janaina Habib Jorge.

**Data curation:** Camilla Olga Tasso.

**Formal analysis:** Túlio Morandin Ferrisse.

**Funding acquisition:** Camilla Olga Tasso.

**Investigation:** Camilla Olga Tasso, Beatriz Ribeiro Ribas.

**Methodology:** Camilla Olga Tasso, Beatriz Ribeiro Ribas, Jonatas Silva de Oliveira.

**Project administration:** Janaina Habib Jorge.

**Software:** Túlio Morandin Ferrisse.

**Supervision:** Janaina Habib Jorge.

**Writing – original draft:** Camilla Olga Tasso, Beatriz Ribeiro Ribas.

**Writing – review & editing:** Janaina Habib Jorge.

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
