## [Decision Letter · Decision Letter 0]

15 May 2024

PONE-D-24-14563The antimicrobial activity of an antiseptic soap against *Candida albicans*and *Streptococcus mutans*single and dual-species biofilms on denture base and reline acrylic resins*****PLOS ONE*

Dear Dr. Jorge,

*Thank you for submitting your manuscript to PLOS ONE. After careful consideration, we feel that it has merit but does not fully meet PLOS ONE’s publication criteria as it currently stands. Therefore, we invite you to submit a revised version of the manuscript that addresses the points raised during the review process.*

*Please submit your revised manuscript by Jun 29 2024 11:59PM. If you will need more time than this to complete your revisions, please reply to this message or contact the journal office at plosone@plos.org. *

*Please include the following items when submitting your revised manuscript:*

*A rebuttal letter that responds to each point raised by the academic editor and reviewer(s). You should upload this letter as a separate file labeled 'Response to Reviewers'.*

*A marked-up copy of your manuscript that highlights changes made to the original version. You should upload this as a separate file labeled 'Revised Manuscript with Track Changes'.*

*An unmarked version of your revised paper without tracked changes. You should upload this as a separate file labeled 'Manuscript'.*

**

*We look forward to receiving your revised manuscript.*

*Kind regards,*

*Geelsu Hwang, Ph.D.*

Academic Editor

*PLOS ONE*

*Journal Requirements:*

"The authors thank the Coordination of Superior Level Staff Improvement (CAPES) for the financial support."

4. Please note that your Data Availability Statement is currently missing the repository name, the DOI/accession number of each dataset or a direct link to access each database. If your manuscript is accepted for publication, you will be asked to provide these details on a very short timeline. We therefore suggest that you provide this information now, though we will not hold up the peer review process if you are unable.

5. Please amend either the title on the online submission form (via Edit Submission) or the title in the manuscript so that they are identical.

**

Reviewers' comments:

*Reviewer's Responses to Questions*

*

**Comments to the Author**
*

1. Is the manuscript technically sound, and do the data support the conclusions?

*The manuscript must describe a technically sound piece of scientific research with data that supports the conclusions. Experiments must have been conducted rigorously, with appropriate controls, replication, and sample sizes. The conclusions must be drawn appropriately based on the data presented. *

*Reviewer #1: Yes*

*Reviewer #2: Partly*

*Reviewer #3: No*

*Reviewer #4: Partly*

*2. Has the statistical analysis been performed appropriately and rigorously? *

*Reviewer #1: Yes*

*Reviewer #2: No*

*Reviewer #3: No*

*Reviewer #4: Yes*

*3. Have the authors made all data underlying the findings in their manuscript fully available?*

*The PLOS Data policy requires authors to make all data underlying the findings described in their manuscript fully available without restriction, with rare exception (please refer to the Data Availability Statement in the manuscript PDF file). The data should be provided as part of the manuscript or its supporting information, or deposited to a public repository. For example, in addition to summary statistics, the data points behind means, medians and variance measures should be available. If there are restrictions on publicly sharing data—e.g. participant privacy or use of data from a third party—those must be specified.*

*Reviewer #1: Yes*

*Reviewer #2: Yes*

*Reviewer #3: No*

*Reviewer #4: Yes*

*4. Is the manuscript presented in an intelligible fashion and written in standard English?*

*PLOS ONE does not copyedit accepted manuscripts, so the language in submitted articles must be clear, correct, and unambiguous. Any typographical or grammatical errors should be corrected at revision, so please note any specific errors here.*

*Reviewer #1: No*

*Reviewer #2: Yes*

*Reviewer #3: No*

*Reviewer #4: Yes*

*5. Review Comments to the Author*

*Please use the space provided to explain your answers to the questions above. You may also include additional comments for the author, including concerns about dual publication, research ethics, or publication ethics. (Please upload your review as an attachment if it exceeds 20,000 characters)*

*Reviewer #1: According to the authors, this study aimed to evaluate the effect of an antiseptic soap on single and dual-species biofilms of C. albicans and S. mutans on denture base and reline resins. As the authors have articulated, denture stomatitis is a disease that affects most denture users, and accessible and low-cost approaches to treat and prevent it are still necessary. There is merit to this paper, however, I think some minor revisions should be performed.*

1) There are some minor language issues (for instance, line 31, page 2: “overnight” does not need to be written in italics; line 92, page 4: I think the authors meant “wearers”, not “wears”). Also, there are multiple words across the manuscript that should be separated but are not.

2) I suggest lines 121-124 from the Experimental groups topic should be moved to the Prevention protocol topic. I think this part of the manuscript is a little confusing. Perhaps a flowchart would improve the understanding of the experiments.

3) I would also suggest lines 125-126 from the Experimental groups topic should be moved to the Disinfection protocol topic.

4) The authors should provide more information regarding the strains used in this study.

5) In my opinion, there are too many figures in this manuscript. I suggest the authors should group them in order to reduce their number. For example, instead of having four different figures for the same experiment, the authors should group them in one, and display them as A-D.

*Nonetheless, this is a good paper that tackles a relevant health issue and provides a simple and affordable solution.*

*Reviewer #2: Methods*

• It is important to clearly explain how the sample size was determined or decided upon.

• What is the reason that the experiment was not planned to be conducted in triplicate?

• According to Ref 14,18, the author should briefly describe how the author prepared soap solutions. What concentration was used in this study?

• For prevention protocol, the samples were immersed in the solutions for 7, 14, and 28 days. It is unclear whether the solution was changed every day, or how the specimens were treated?

• For statistical test, did the author conduct post-hoc adjustment for multiple comparisons?

Results

• There are too many figures. Some results can be displayed in one figure, such as figure 1-2 prevention protocol: Denture acrylic base and relining material, or in table format, such as fluorescence values.

• All figures have low resolution. It lacks sharpness.

Discussion

• The first paragraph of the discussion is redundant with the introduction. The author should highlight the important findings of this study.

• Regarding preventative protocol, what is the influence of different immersion durations (7, 14, 28) on the prevention of biofilm formation? Does the result provide evidence for the feasibility of clinical application? What is the underlying mechanism of this protocol that causes the active substance to remain on the surface of the denture?

• Why is the reduction of biofilm in relining materials (2-log) lower than that in denture acrylic resin (3-log)? Is there any reason for the sort of material?

Miscellaneous

Line 422: spacing “thepotential”

Line 331: spacing “wasobserved”

Line 332: spacing “inrelation”

Line 418: spacing “theefficacy”

*Line 422: spacing “thepotential”*

*Reviewer #3: In fact, the manuscript addresses important concerns about the need for an alternative disinfection protocol for denture. Although, it reports biofilm control in health care, the manuscript is fragile due to some reasons: I) changes in the physical properties of reline and acrylic resins, as well as the corrosive nature of the soap, was not addressed; II) the exact mechanism of action was not clarified; III) The effects of biological variation among strains of the same species were not considered by the authors, given that only one strain was tested.*

Methods

What was the reasonable determine the Prevention protocol? The authors reported that the samples were immersed in the solutions for 7, 14 and 28 days before he single and dual-species biofilms formation. The author pointed out that the samples remained in the solutions for 8 hours and in distilled water for the other 16 hours, which simulated the period of use of the denture. Did the authors evaluated if the tested materials can efficiently absorb the hygiene solutions? Was a controlled release test carried out?

In view of distinct morphology of yeasts (yeast and hyphae), grid slides offer a more direct, precise, and informative method for determining yeast concentration, instead of absorbance measurements.

Was the application of the disinfection protocol carried out in the same well-plate that the biofilm was grown? Why? Would the remaining planktonic cells, on specimens and plate surfaces, impact the viable cell enumeration?

After implementing both prevention and disinfection protocols, the remaining biofilm was scraped off using a pipette tip. The authors prepared specimens with a surface roughness value ranging between 2.7 and 3.7 μm. Such surface roughness harbors valleys that may not be reached by a pipette tip. How can the authors be certain that all remaining biofilm was completely removed and accurately quantified?

BHI medium also promotes the growth of Candida albicans? In the dual species biofilm, how did the authors prevent / reduce C. albicans growth or selectively count Streptococcus mutans colonies?

Do the authors consider that clinical or multidrug resistant strains would have the same behavior?

The authors should indicate the number of technical and biological replicates of the study.

With respect to statistical analysis, considering that immersion in sodium hypochlorite reduced CFU counts and metabolic activity to zero, the group should not be included in further analyses due to its lack of variability and distinguishable outcomes. Therefore, I suggest that author review the statistical design and the description of all results.

Results

Taking into the account all the results, the figures could be summarized in a maximum 3 or 4 images. For instance, all the results referring to CFU/counts and metabolic activity could be showed in tables, with the specific p values for the pairwise comparisons.

From imagens 1 to 4, the authors indicated that some groups presented reduction in microbial viability. I noticed a slight reduction. Is 1-log reduction considered of paramount importance in this field?

How can the authors explain a reduction in cell metabolism in an almost similar biofilm rates, measured by CFU counts?

With respect to disinfection protocol, I suggest that authors report their findings in terms of biofilm viability instead of biofilm formation.

Discussion

While the authors suggest that making a direct comparison with other results is impossible and emphasize the importance of the present study, I propose that other points should be discussed. For instance, considering the mechanism of action of the soap in both protocols could provide valuable insights. Could it effectively penetrate the deeper layers of the biofilm by disrupting the extracellular polymeric substance (EPS) matrix? Moreover, in the prevention protocol, is there evidence to suggest that it can adsorb to the surface of materials, potentially hindering biofilm formation? In this study only the chemical activity was addressed. And about the mechanical biofilm removal? Could it improve the antibiofilm activity of the soap?

The authors could have evaluated the effects of the hygiene protocols on other central variables, such as biofilm removal ability and extracellular matrix, that are together important virulence attributes. Additional microscopy analyses would increase the study's impact and contribute to a better interpretation of the results obtained. Ideally, an antimicrobial agent should reduce the cell viability and remove the biofilm, since the residual bioincrustation favors the increase in biomass and thickness and subsequent recolonization of the surface.

Minor comments

Digitation typographical errors:

Line 47: “poor hygieneof the”

Line 168: “incubationperiod”

Line 310: “propertyalterations”

Line 331: “it wasobserved”

Line 332: “inrelation”

Line 361: “reducethe”

Line 368: “Lifebuoyliquid”

The author reported the proposition of an effective protocols for the treatment and prevention of denture stomatitis. Indeed, they evaluated the effectiveness of hygiene protocols on both preventing and removing biofilm on the surface of denture materials. The effect of the protocols on denture stomatitis is far from the findings presented here.

Why was “Overnight” italicized throughout the manuscript?

In particular, within the discussion section, references are presented using both author/year format and a numerical list.

Attention should be driven to the description of the bacteria and yeasts genus and species. After a scientific name is written in full, in a second occurrence in the text it is mandatory to abbreviate the genus name by just using first initial and then a period to represent the genus. This recommendation is addressed to the abstract section and to the body of the manuscript as well. Both latin expressions and scientific names should be italicized.

*Figure legends should be placed after the references, should briefly descript the content or purpose of the figure, the statistical test, and any relevant details necessary for interpretation, such as experimental conditions or scale information.*

*Reviewer #4: This manuscript is very relevant because the findings demonstrate the effectiveness of Lifebuoy soap solution as a cost-effectiveness and easily accessible antimicrobial agent for cleaning complete dentures.*

Abstract

Please replace the term divided to distributed on the phrase “Samples of the resins were divided into groups and add the dimensions of samples (14 mm x 1,2 mm) on line 28.

Please add the Dunnett’s T3 multiple comparisons test on Abstract that had been mentioned on 286 line of Results.

Introduction

Please replace the references 16 and 17 on line 74 from authors who have not studied the effect of sodium hypochlorite on metallic structures with others that demonstrate these findings.

Material and Method

Line 117 -please add the size of aluminum oxide and manufacturer.

Please replace the term divided to distributed on the phrase “The denture base acrylic resin and hard chairside relining samples were divided into experimental groups....on 114 line.

Please add the Dunnett’s T3 multiple comparisons test on the Statistical analysis, that had been mentioned on 286 line of Results.

Discussion

Please rewrite the third paragraph about the prevention protocol (330-337) which is not in accordance with what was reported in the results of the denture relining samples and S. mutans.

Conclusion

I think that your statement just above the Conclusion section regarding that “These findings support the effectiveness of Lifebuoy soap as an antimicrobial agent and highlight its cost-effectiveness and accessibility as a viable option for cleaning partial and complete removable dentures” needs to be changed only for cleaning complete dentures because you didn’t evaluate this antimicrobial action on metallic surface of Removable Partial Dentures.

Figures

Figure 6 - Please correct the acronym of sodium hypochlorite (SD) on X axis.

*Figure 16 - Please keep the same sequence of other figures on X axis (LS, PBS, SH).*

*6. PLOS authors have the option to publish the peer review history of their article (what does this mean?). If published, this will include your full peer review and any attached files.*

**

*Reviewer #1: No*

*Reviewer #2: No*

*Reviewer #3: **Yes: **Viviane de Cássia Oliveira*

*Reviewer #4: No*

**

*While revising your submission, please upload your figure files to the Preflight Analysis and Conversion Engine (PACE) digital diagnostic tool, https://pacev2.apexcovantage.com/. PACE helps ensure that figures meet PLOS requirements. To use PACE, you must first register as a user. Registration is free. Then, login and navigate to the UPLOAD tab, where you will find detailed instructions on how to use the tool. If you encounter any issues or have any questions when using PACE, please email PLOS at figures@plos.org. Please note that Supporting Information files do not need this step.*

---

## [Author Response · Author response to Decision Letter 0]

5 Jun 2024

May 16, 2024.

Dear Editor, PLOS ONE

Dear Professor 

Thank you for your considerations on the manuscript “PONE-D-24-14563 The antimicrobial activity of an antiseptic soap against Candida albicansand Streptococcus mutanssingle and dual-species biofilms on denture base and reline acrylic resins”. We are submitting the revised manuscript. We would like to affirm our great interest in publishing the article in this respected journal and we believe that this new version of the manuscript is now suitable for publication.

Reviewer #1: 

Firstly, we would like to thank you for your considerations on the manuscript for the overall improvement of our study. 

Reviewer #1: According to the authors, this study aimed to evaluate the effect of an antiseptic soap on single and dual-species biofilms of C. albicans and S. mutans on denture base and reline resins. As the authors have articulated, denture stomatitis is a disease that affects most denture users, and accessible and low-cost approaches to treat and prevent it are still necessary. There is merit to this paper, however, I think some minor revisions should be performed.

1) There are some minor language issues (for instance, line 31, page 2: “overnight” does not need to be written in italics; line 92, page 4: I think the authors meant “wearers”, not “wears”). Also, there are multiple words across the manuscript that should be separated but are not.

Response: The minor language issues and digitation typographical errors were corrected.

2) I suggest lines 121-124 from the Experimental groups topic should be moved to the Prevention protocol topic. I think this part of the manuscript is a little confusing. Perhaps a flowchart would improve the understanding of the experiments.

Response: The Flowcharts 1 and 2 were prepared to improve the understanding of the experiments as suggested. 

3) I would also suggest lines 125-126 from the Experimental groups topic should be moved to the Disinfection protocol topic.

Response: Changes to the text were made to improve the understanding of the experiments as suggested. 

4) The authors should provide more information regarding the strains used in this study.

Response: The information was provided as suggested (C. albicans ATCC 90028 and S. mutans UA 159).

5) In my opinion, there are too many figures in this manuscript. I suggest the authors should group them in order to reduce their number. For example, instead of having four different figures for the same experiment, the authors should group them in one, and display them as A-D.

Response: The Figures were group them in order to reduce their number as suggested.

Nonetheless, this is a good paper that tackles a relevant health issue and provides a simple and affordable solution.

We would like to thank you again for your considerations.

Reviewer #2: 

Firstly, we would like to thank you for your considerations on the manuscript for the overall improvement of our study. 

Reviewer #2: 

Methods

• It is important to clearly explain how the sample size was determined or decided upon.

Response: This information was added in the Methods: “The n value was estimated based on previous studies [19-24].”

• What is the reason that the experiment was not planned to be conducted in triplicate?

Response: This information was added in the Methods: “The experiments were performed in triplicate, on three different occasions (n = 9).”

• According to Ref 14,18, the author should briefly describe how the author prepared soap solutions. What concentration was used in this study?

Response: The information was added as suggested.

• For prevention protocol, the samples were immersed in the solutions for 7, 14, and 28 days. It is unclear whether the solution was changed every day, or how the specimens were treated?

Response: This information was added in the Methods: “The solutions were changed every day. Overnight denture disinfection was simulated. Thus, the samples remained in the solutions for 8 hours and in distilled water for the other 16 hours, which simulated the period of use of use of the denture.” 

• For statistical test, did the author conduct post-hoc adjustment for multiple comparisons?

Response: Thank you for your note. The post-hoc adjustment for multiple comparisons was made only for Alamar blue assay in disinfection protocol (Dunnett’s T3 and Dunn’ multiple comparisons tests). For all other statistical analysis, the multiple comparisons were performed by mean ± 95% confidence interval estimation and the followed results illustrated in graphs. In this case, if the space between the error bar did not match there is a significant statistical difference among the groups evaluated. This explanation was added in the manuscript methodology.

Results

• There are too many figures. Some results can be displayed in one figure, such as figure 1-2 prevention protocol: Denture acrylic base and relining material, or in table format, such as fluorescence values.

Response: The Figures were group them in order to reduce their number as suggested.

• All figures have low resolution. It lacks sharpness.

Response: The Figures were improved. 

Discussion

• The first paragraph of the discussion is redundant with the introduction. The author should highlight the important findings of this study.

Response: This sentence was added in order to highlight the important findings of this study; “The results indicate an easy and inexpensive method for denture disinfection, particularly for elderly patients with limited manual dexterity. In addition, immersion in disinfectant solutions could be a common protocol in hospitals and nursing homes for denture care among institutionalized patients.”

• Regarding preventative protocol, what is the influence of different immersion durations (7, 14, 28) on the prevention of biofilm formation? Does the result provide evidence for the feasibility of clinical application? What is the underlying mechanism of this protocol that causes the active substance to remain on the surface of the denture?

Response: This sentence was added in the Discussion: “To simulate clinical conditions, various immersion times were evaluated. This is important because the daily immersion of the dentures in disinfectant solutions can cause the incorporation of substances into the acrylic resin, which can cause superficial changes and, consequently, influence the process of adhesion and biofilm formation. Due to the daily exchange of solutions, degradation conditions and release of components can be maintained over time. In this study, there was minimal impact of immersion time on biofilm adhesion and formation.”

• Why is the reduction of biofilm in relining materials (2-log) lower than that in denture acrylic resin (3-log)? Is there any reason for the sort of material?

Response: This sentence was added in the Discussion: “This difference between the resins can be attributed to the material's surface characteristics. Relining materials, with their greater surface roughness compared to denture base resins, create a more favorable environment for biofilm accumulation [24].”

Miscellaneous

Line 422: spacing “thepotential”

Line 331: spacing “wasobserved”

Line 332: spacing “inrelation”

Line 418: spacing “theefficacy”

Line 422: spacing “thepotential”

Response: The digitation typographical errors were corrected.

Reviewer #3: 

Firstly, we would like to thank you for your considerations on the manuscript for the overall improvement of our study. 

Reviewer #3: In fact, the manuscript addresses important concerns about the need for an alternative disinfection protocol for denture. Although, it reports biofilm control in health care, the manuscript is fragile due to some reasons: 

I) changes in the physical properties of reline and acrylic resins, as well as the corrosive nature of the soap, was not addressed;

Response: This sentence was added in the text: “The soap solution can be an alternative for cleaning removable dentures, considering its effectiveness in reducing biofilm and the absence of cytotoxicity, as well as the unaltered physical and mechanical properties of the acrylic resins after immersion in this solution [14, 18-21]. In additional, it had a good acceptance by denture wearers [18].”

II) the exact mechanism of action was not clarified; 

Response: The mechanism of action will be evaluated in future studies and this information was added in the discussion. However, a possible mechanism of action was added: “In addition, the soap's mechanisms of action may also be related to the presence of benzalkonium chloride, a quaternary ammonium compound common in non-alcohol hand sanitizers [44]. The positively charged "headgroup" of benzalkonium chloride is attracted to and progressively adsorbs to the negatively charged phosphate heads of phospholipids in the lipid bilayer. This accumulation disrupts the membrane's fluidity, consequently creating hydrophilic gaps within the structure [45]. In addition, the alkyl chain "tail" component of benzalkonium chloride disorganizes the membrane bilayer by inserting itself into the barrier and disrupting its physical and biochemical properties [44,45].”

III) The effects of biological variation among strains of the same species were not considered by the authors, given that only one strain was tested.

Response: The effects of biological variation among strains of the same species will be evaluated in future studies and this information was added in the discussion.

Methods

What was the reasonable determine the Prevention protocol? The authors reported that the samples were immersed in the solutions for 7, 14 and 28 days before he single and dual-species biofilms formation. The author pointed out that the samples remained in the solutions for 8 hours and in distilled water for the other 16 hours, which simulated the period of use of the denture. Did the authors evaluated if the tested materials can efficiently absorb the hygiene solutions? Was a controlled release test carried out?

Response: This sentence was added in the Discussion: “To simulate clinical conditions, various immersion times were evaluated. This is important because the daily immersion of the dentures in disinfectant solutions can cause the incorporation of substances into the acrylic resin, which can cause superficial changes and, consequently, influence the process of adhesion and biofilm formation [12,14]. Due to the daily exchange of solutions, degradation conditions and release of components can be maintained over time.” 

In view of distinct morphology of yeasts (yeast and hyphae), grid slides offer a more direct, precise, and informative method for determining yeast concentration, instead of absorbance measurements.

Was the application of the disinfection protocol carried out in the same well-plate that the biofilm was grown? Why? Would the remaining planktonic cells, on specimens and plate surfaces, impact the viable cell enumeration?

Response: In order to avoid this problem, each well was washed twice with PBS, and the specimens were transferred to a new 24-well plate before the assays. This information was added in the in the Methods.

After implementing both prevention and disinfection protocols, the remaining biofilm was scraped off using a pipette tip. The authors prepared specimens with a surface roughness value ranging between 2.7 and 3.7 μm. Such surface roughness harbors valleys that may not be reached by a pipette tip. How can the authors be certain that all remaining biofilm was completely removed and accurately quantified?

Response: The methodology was based in previous studies and the references were added in the sentence. In addition, the positive and negative control group was used and the experiments were performed by a single experienced operator to ensure a standard protocol. This information was added in the Methods.

BHI medium also promotes the growth of Candida albicans? In the dual species biofilm, how did the authors prevent / reduce C. albicans growth or selectively count Streptococcus mutans colonies?

Response: In the dual-species biofilm, BHI medium with Amphotericin B was used in order to grow only Streptococcus mutans. This information was added in the Methodology. 

Do the authors consider that clinical or multidrug resistant strains would have the same behavior?

Response: The sentence was added: “Further studies with more complex biofilms and studies on possible mechanisms of action are necessary to substantiate the efficacy of the proposed protocol.”

The authors should indicate the number of technical and biological replicates of the study.

Response: The sentence was added as suggested: “The experiments were performed in triplicate, on three different occasions (n = 9).”

With respect to statistical analysis, considering that immersion in sodium hypochlorite reduced CFU counts and metabolic activity to zero, the group should not be included in further analyses due to its lack of variability and distinguishable outcomes. Therefore, I suggest that author review the statistical design and the description of all results.

Response: Thank you for your note. If the treatment protocols reached out “zero-values”, the values of the following group have not been added to conduct the statistical test. In addition, this values were added only in graph to illustrated the results and it was considered p-values (p < 0.05) comparison of other groups in which did not present “zero-values”.

Results

Taking into the account all the results, the figures could be summarized in a maximum 3 or 4 images. For instance, all the results referring to CFU/counts and metabolic activity could be showed in tables, with the specific p values for the pairwise comparisons.

Response: The Figures were group them in order to reduce their number as suggested.

From imagens 1 to 4, the authors indicated that some groups presented reduction in microbial viability. I noticed a slight reduction. Is 1-log reduction considered of paramount importance in this field?

Response: According to the CEN standard EN 1657:2007, for a sanitizer to be considered effective against fungi, it must achieve a 3-log reduction in the number of fungal colonies compared to the positive control (disinfection protocol). Figures 1 to 4 show the results of the prevention protocol and a 1-log reduction was significant in reducing biofilm formation. Additional information was added in the Discussion to clarify the results.

How can the authors explain a reduction in cell metabolism in an almost similar biofilm rates, measured by CFU counts?

Response: This sentence was added in the Discussion: “The Alamar Blue® test showed a reduction in cellular metabolism for the experimental groups in relation to the negative control, corroborating the results of the counting colony-forming units. Thereby, the disinfectant solutions promoted, in addition to reducing biofilm formed on the resins, a reduction in their metabolism.”

With respect to disinfection protocol, I suggest that authors report their findings in terms of biofilm viability instead of biofilm formation.

Response: The changes were made as suggested.

Discussion

While the authors suggest that making a direct comparison with other results is impossible and emphasize the importance of the present study, I propose that other points should be discussed. For instance, considering the mechanism of action of the soap in both protocols could provide valuable insights. Could it effectively penetrate the deeper layers of the biofilm by disrupting the extracellular polymeric substance (EPS) matrix? Moreover, in the prevention protocol, is there evidence to suggest that it can adsorb to the surface of materials, potentially hindering biofilm formation? 

Response: Information regarding these aspects was added in the Discussion.

In this study only the chemical activity was addressed. And about the mechanical biofilm removal? Could it improve the antibiofilm activity of the soap?

Response: This sentence was added in the Discussion: “It is important to emphasize that brushing remains a cornerstone of oral hygiene, and the proposed protocol could offer an additional benefit when used in conjunction with regular brushing. Previous stud

---

## [Decision Letter · Decision Letter 1]

25 Jun 2024

The antimicrobial activity of an antiseptic soap against *Candida albicans*and *Streptococcus mutans*single and dual-species biofilms on denture base and reline acrylic resins****

*PONE-D-24-14563R1*

*Dear Dr. Jorge,*

*We’re pleased to inform you that your manuscript has been judged scientifically suitable for publication and will be formally accepted for publication once it meets all outstanding technical requirements.*

*Within one week, you’ll receive an e-mail detailing the required amendments. When these have been addressed, you’ll receive a formal acceptance letter and your manuscript will be scheduled for publication.*

*An invoice will be generated when your article is formally accepted. Please note, if your institution has a publishing partnership with PLOS and your article meets the relevant criteria, all or part of your publication costs will be covered. Please make sure your user information is up-to-date by logging into Editorial Manager at Editorial Manager® and clicking the ‘Update My Information' link at the top of the page. If you have any questions relating to publication charges, please contact our Author Billing department directly at authorbilling@plos.org.*

*If your institution or institutions have a press office, please notify them about your upcoming paper to help maximize its impact. If they’ll be preparing press materials, please inform our press team as soon as possible -- no later than 48 hours after receiving the formal acceptance. Your manuscript will remain under strict press embargo until 2 pm Eastern Time on the date of publication. For more information, please contact onepress@plos.org.*

*Kind regards,*

*Geelsu Hwang, Ph.D.*

Academic Editor

*PLOS ONE*

* *

*Reviewers' comments:*

*Reviewer's Responses to Questions*

*

**Comments to the Author**
*

*1. If the authors have adequately addressed your comments raised in a previous round of review and you feel that this manuscript is now acceptable for publication, you may indicate that here to bypass the “Comments to the Author” section, enter your conflict of interest statement in the “Confidential to Editor” section, and submit your "Accept" recommendation.*

*Reviewer #1: All comments have been addressed*

*Reviewer #3: (No Response)*

*Reviewer #4: All comments have been addressed*

*2. Is the manuscript technically sound, and do the data support the conclusions?*

*The manuscript must describe a technically sound piece of scientific research with data that supports the conclusions. Experiments must have been conducted rigorously, with appropriate controls, replication, and sample sizes. The conclusions must be drawn appropriately based on the data presented. *

*Reviewer #1: Yes*

*Reviewer #3: (No Response)*

*Reviewer #4: Yes*

*3. Has the statistical analysis been performed appropriately and rigorously? *

*Reviewer #1: Yes*

*Reviewer #3: (No Response)*

*Reviewer #4: Yes*

*4. Have the authors made all data underlying the findings in their manuscript fully available?*

*The PLOS Data policy requires authors to make all data underlying the findings described in their manuscript fully available without restriction, with rare exception (please refer to the Data Availability Statement in the manuscript PDF file). The data should be provided as part of the manuscript or its supporting information, or deposited to a public repository. For example, in addition to summary statistics, the data points behind means, medians and variance measures should be available. If there are restrictions on publicly sharing data—e.g. participant privacy or use of data from a third party—those must be specified.*

*Reviewer #1: Yes*

*Reviewer #3: (No Response)*

*Reviewer #4: Yes*

*5. Is the manuscript presented in an intelligible fashion and written in standard English?*

*PLOS ONE does not copyedit accepted manuscripts, so the language in submitted articles must be clear, correct, and unambiguous. Any typographical or grammatical errors should be corrected at revision, so please note any specific errors here.*

*Reviewer #1: No*

*Reviewer #3: (No Response)*

*Reviewer #4: Yes*

*6. Review Comments to the Author*

*Please use the space provided to explain your answers to the questions above. You may also include additional comments for the author, including concerns about dual publication, research ethics, or publication ethics. (Please upload your review as an attachment if it exceeds 20,000 characters)*

*Reviewer #1: I have checked the revised manuscript and it has been much improved. The authors have made a great effort to address all of the reviewers' concerns. This manuscript is in a nice condition for acceptance.*

*Reviewer #3: In fact, the authors presented a reviewed version of the manuscript. However, many of the previous concerns were only justified to be addressed in future studies. This approach undermines the current validity and completeness of the manuscript, as crucial issues remain unresolved.*

Mechanism of action was explained based on properties of the membrane bilayer. However, considering microorganisms, it is important to note that the cleanser must first penetrate a complex cell wall structure before reaching the membrane bilayer. This adds an additional layer of complexity.

*The manuscript appears to offer only incremental insights rather than introducing significant new findings or perspectives to the field. The existing body of literature already covers much of the content presented in the manuscript, thereby limiting its potential to substantially advance the field.*

*Reviewer #4: (No Response)*

*7. PLOS authors have the option to publish the peer review history of their article (what does this mean?). If published, this will include your full peer review and any attached files.*

**

*Reviewer #1: No*

*Reviewer #3: No*

*Reviewer #4: No*

---

## [Editor Report · Acceptance letter]

2 Jul 2024

PONE-D-24-14563R1 

PLOS ONE

Dear Dr. Jorge, 

I'm pleased to inform you that your manuscript has been deemed suitable for publication in PLOS ONE. Congratulations! Your manuscript is now being handed over to our production team.

Kind regards, 

on behalf of

Dr. Geelsu Hwang 

Academic Editor

PLOS ONE